# Evaluation of Pre-Applied Conductive Materials in Electrode Grids for Longterm EEG Recording

**DOI:** 10.3390/s25226810

**Published:** 2025-11-07

**Authors:** Carlos F. da Silva Souto, Wiebke Pätzold, Joanna E. M. Scanlon, Axel H. Winneke, Stefan Debener, Karen Insa Wolf

**Affiliations:** 1Fraunhofer Institute for Digital Media Technology IDMT, Branch Hearing, Speech and Audio Technology HSA, 26129 Oldenburg, Germany; carlos.filipe.da.silva.souto@idmt.fraunhofer.de (C.F.d.S.S.); wiebke.paetzold@idmt.fraunhofer.de (W.P.); joanna.scanlon@idmt.fraunhofer.de (J.E.M.S.); axel.winneke@idmt.fraunhofer.de (A.H.W.); stefan.debener@uni-oldenburg.de (S.D.); 2Neuropsychology Lab, Department of Psychology, University of Oldenburg, 26129 Oldenburg, Germany; 3Cluster of Excellence “Hearing4all”, University of Oldenburg, 26129 Oldenburg, Germany

**Keywords:** flex-printed electrodes, trEEGrid, hydrogel, silicone gel, long-term EEG, mobile EEG

## Abstract

Most long-term mobile EEG monitoring systems require professional application of the electrodes, which makes them inconvenient for everyday use. Additionally, many materials that facilitate EEG application, such as dry electrodes, may cause discomfort when worn for longer periods of time. To address these problems, we designed flex-printed EEG electrode grids (trEEGrid) and evaluated signal quality based on two pre-applied conductive materials. Self-applicable trEEGrid patches with a conductive solid hydrogel and a novel silicone-based dry material were used in a day-long (5–6 h) recording session, which included a 4 h continuous recording of impedance levels, as well as two auditory task recordings in the morning and afternoon. The signal-to-noise ratio (SNR) of the auditory evoked potentials (AEPs), AEP morphology, and impedance levels of the conductive materials were compared to evaluate overall signal quality, and further comparisons took place between the morning and afternoon sessions to evaluate signal deterioration over time. Comparable impedance values were observed for both silicone and hydrogel materials, but the silicone material exhibited a higher outlier rate, with impedance values over 200 kΩ. Over time, the impedance values increased for the silicone material and decreased for the hydrogel material. The morphology of the AEP was reproduced comparably well with both materials, with reasonable SNRs in both the morning and the afternoon. In conclusion, when combined with flex-printed electrode grids, silicone and hydrogel materials make it feasible to collect high-quality long-term EEG signals with high wearing comfort.

## 1. Introduction

Electroencephalography (EEG) is widely used in healthcare and neuropsychological research to record brain activity and study brain function. Several health applications—such as home-based sleep monitoring, epileptic activity tracking in daily life, and driver state assessment—remain difficult to address with existing systems due to limitations in comfort and long-term wearability. For these mobile, long-duration applications, EEG sensors are required to provide both high signal quality and a high level of user comfort. In addition, depending on the context, it is advantageous for users to be able to apply the electrode system themselves.

Conventional medical-grade electrodes, such as gel-based Ag/AgCl sintered ring electrodes, provide high signal quality but are not well suited for long-term use. Conductive gels bridging skin and the electrode surface can dry out over time, leading to signal impedance deterioration and signal degradation. Moreover, rigid metal electrodes and the attached wires are inconvenient for extended wear and their application typically requires trained personnel, which limits their use outside controlled environments such as hospitals and laboratories. Dry electrode systems have been proposed as a more user-friendly alternative, as they eliminate the need for gel and can often be worn after a minimal preparation time [e.g., [1]]. However, rigid and flexible dry electrodes require contact pressure to maintain electrode–skin coupling, which restricts long-term wear due to headaches, discomfort, and pinching sensations that may arise within an hour of use [2]. In addition, depending on their design, dry electrodes may suffer from elevated impedance values, which in turn increases noise levels [3,4], as well as the susceptibility of EEG signals to motion artifacts caused by shifts in the system relative to the skin [5,6].

Electrode systems such as cEEGrids [7,8] are based on lightweight, flex-printed circuit boards (flex-PCBs) that adhere directly to the skin without the need for additional pressure on the head. These around-the-ear EEG systems have demonstrated significant advantages for long-term recordings, offering both comfort and stable signal quality [7]. Their use, however, is limited to near-hairless skin regions such as the forehead, temples, or areas around the ears. Despite this restriction, flex-printed electrodes have proven effective for capturing key EEG signal features such as alpha oscillations, auditory-evoked potentials and event-related potentials [7,8,9], and they can capture epileptiform activity and sleep EEG signatures [8,10,11]. For example, da Silva Souto and colleagues [10] demonstrated the feasibility of using flex-printed electrodes for overnight sleep EEG recordings, systematically investigating electrode placements around the ear and in the facial region. Additionally, Scanlon et al. [12] demonstrated that frontal and centroparietal spectral activity could be equivalently captured using a cap and a flex-printed electrode grid.

Nevertheless, conventional flex-printed electrode grids such as the cEEGrid rely on electrolyte gel, which limits their ease of use and reduces self-applicability. To progress towards a fully self-applicable and user-friendly EEG solution, naïve users should be able to establish stable electrode–skin connections without external assistance or extensive training. Souto et al. [13] addressed this challenge by employing pre-gelled neonatal ECG electrodes in a grid configuration [trEEGrid] that allowed for self-application. Building on this concept, the trEEGrid prototype has since been refined into a flex-printed electrode patch in which conductive material is pre-applied to the electrode sites. This advancement eliminates the need for separate gel application, simplifies handling, and facilitates reliable self-application, as illustrated in Figure 1.

The current study builds on previous work using trEEGrids [12,13] by demonstrating the ability to record evoked responses and impedance values in order to compare signal quality between sensors made of two conductive materials. In this study, we present and evaluate two conductive materials with respect to their suitability for long-term EEG recordings: one is a curing conductive hydrogel and the other one a novel silicone-based material. Signal quality was assessed during EEG recording sessions lasting five to six hours. In addition to the monitoring of electrode impedance dynamics, we evaluated the signal-to-noise ratio (SNR) of auditory-evoked potentials (AEPs). This is the first study in which the flex-printed trEEGrid sensor system was used in conjunction with a pre-applied conductive material. Previous trEEGrid studies used either small, neonatal ECG electrodes arranged in the trEEGrid layout [12] or the standard conductive viscous EEG gel, which was applied onto the trEEGrid just before placing the grid onto the participant [12].

## 2. Materials and Methods

### 2.1. Materials

The layout and placement of the electrodes in the trEEGrid electrode patch are illustrated in Figure 2. A mirrored design allows for parallel recordings to be made on the same individual using both conductive materials. The design includes eight recording electrodes and two more electrodes serving as reference and ground. Three electrodes (reference, ground, and one recording electrode) are positioned behind the ear, while the remaining electrodes are placed in the facial area. Electrodes 1, 2, and 3 enable, in addition to EEG recordings, the derivation of horizontal, vertical, and diagonal electrooculogram (EOG) signals, similar to EOG recordings in sleep laboratories (cf. [13]). Electrodes 7 and 8 aim to capture facial electromyogram (EMG) signals, comparable to EMG recordings in sleep laboratories. EOG- and EMG-specific derivations will not be discussed further in this paper, as the focus is on EEG data. Two conductive materials were applied to the electrode areas of the flexible grid: a curing conductive solid gel (hydrogel) on the left side of the face, and a novel silicone-based dry material on the right side, as illustrated in Figure 2.

Hydrogel made from water, a polymer, and sodium chloride is commonly employed for electrocardiogram recordings (MTG Imiella Medizintechnik, Lugau, Germany). It was manually applied to the electrode surface in liquid form and subsequently cured under UV light to achieve an elastic yet stable and solid droplet shape. The electrode surface was designed as a circle to accommodate the droplet shape of the conductive material. Electrodes with this material were stored in sealed bags prior to use to prevent the hydrogel from drying out. The second material is a novel silicone-based conductive material (DuPont™ Liveo™ Soft Skin Conductive Tape, DuPont, Braine-l’Alleud, Belgium). These electrodes were designed with an oval shape to allow the patch to remain as narrow and delicate as possible. In both cases, the effective electrode surface corresponds to a circle with a 9 mm diameter. Both materials are shown in Figure 3. Both types of electrodes exhibit slight natural adhesion to the skin. Additionally, an adhesive surrounding each electrode is applied to enhance attachment. The self-application process is demonstrated in an online video [14]. Hydrogel material has the natural tendency to lose moisture and dry out over time and when exposed to air. This, in turn, affects its conductive property and consequently the signal. Although we mitigated this problem by using circular adhesives around the hydrogel material to seal it off, loss of moisture cannot be ruled out as we did not specifically measure the water content of the hydrogel material before and after the measurements.

### 2.2. Recording Setup

Each trEEGrid patch was connected via cable to a wireless EEG amplifier (Smarting Sleep, mBrainTrain, Belgrade, Serbia), which was worn in a pouch on a shoulder strap. Data were recorded at a sampling rate of 250 Hz and transmitted via Bluetooth to a Raspberry Pi–based recording station. Synchronization of both recording streams was achieved using the Lab Streaming Layer framework [15].

### 2.3. Procedure

Twenty-four participants were recruited through the University of Oldenburg website and the Fraunhofer IDMT blackboard to take part in the study. Each participant was asked to attend for approximately seven hours, from 10 a.m. to 5 p.m., to complete the entire experimental protocol (see Protocol). Five participants were excluded from the subsequent data analysis for the following reasons: not following the preparation protocol (1); loss of wireless connection during data collection (4). The final dataset included 19 participants (median age = 25 years; age range = 21–61 years; 10 women and 9 men). Participants had no history of neurological diseases or medical implants and had normal hearing and vision, or corrected vision. Written consent was provided by all participants. The protocols were approved by the local ethics board at Carl von Ossietzky University in Oldenburg (reference number 2023-015).

### 2.4. Protocol

Before the experiment began, the participants were sent information on how to prepare, including a copy of the consent form. They were instructed to get plenty of rest before the experiment, to avoid wearing make-up or excessive skin products, and, if applicable, to wear glasses instead of contact lenses to reduce the number of times they blinked. Participants who regularly shave their face were asked to do so on the day of the experiment, since some of the electrodes would be placed on areas of the face with facial hair. Participants were also asked to bring a quiet activity to occupy them during periods without experimental tasks (e.g., a book or laptop), but were asked to avoid using active Bluetooth devices.

Upon arrival, participants reviewed and signed the consent form. They were then provided with an additional briefing about the study (see Figure 4). Each participant took part in two auditory tasks and two visual oddball tasks (one of each in the morning and one in the afternoon), as well as a long-term task containing movement and rest conditions. The current study presents data from the two auditory tasks (AEP response) as well as impedance data for all tasks concatonated together from the morning until the afternoon, including the long-term measurement. Task-specific data from other tasks, including the visual oddball task, and eyes-open and -closed resting tasks, may be presented in subsequent publications.

All measurements were taken in an ordinary office, so all the typical electromagnetic influences of daily office life were present. To compare the suitability of the test material for long-term EEG measurements, this paper focuses solely on the results of the two auditory tasks and the continuous impedance measurements throughout the day.

Afterwards, the participant’s skin was prepared for the patch application. This process began with the skin being lightly abraded using Abralyt HiCl (EasyCap GmbH, Wörthsee, Germany) gel and cotton pads in three circular motions on each expected electrode position. Residual gel was removed with water and a towel before these skin positions were cleaned with alcohol; after the alcohol evaporated, the patches were applied. To directly compare both materials, the hydrogel patch was applied on the left side of the head and the silicone patch was placed on the right side. The patches were connected to separate amplifiers, which were placed in a shoulder strap. Amplifiers were wirelessly connected to Raspberry Pi EEG acquisition units via Bluetooth.

Participants then performed an auditory task, followed by a visual oddball task after a short break. This was followed by a continuous EEG measurement, which lasted approximately four hours or until 4 p.m. During this period, the participants completed three short tasks, after which they spent the rest of the afternoon performing quiet activities in the experimental room. They performed the same three short tasks again at the end of the recording period. For the short tasks, participants first lay on the day bed and stared at a fixation cross on the ceiling for three minutes with their eyes open. This was followed by a further three minutes in the same position with their eyes closed. For the third task, participants performed walking movements on a stationary exercise stepper for three minutes. The stepper rotates with each step to create movement patterns mimicking walking movements. During the continuous recording phase, participants could leave the room briefly to use the toilet or purchase lunch from a local bistro. However, they were asked not to leave the building. At the end of the experiment, the three 3 min tasks were performed again in the same order (lying down with eyes open and closed, then using the stepper), after which the continuous EEG recording phase ended. Finally, participants performed the auditory and visual tasks again. It was approximately 4:30 p.m. at this point, and participants were provided with a brief debriefing before going home.

#### 2.4.1. Auditory Task

AEPs are well-known, robust brain signals in response to audible sounds [16,17] and have previously been used to evaluate EEG signals and setups [18,19]. The very early (0–8 ms) and subsequent (8–50 ms) components of the AEP tend to be stimulus-dominated and are not greatly affected by listener states such as arousal or attention levels [20,21]. Late cortical AEP components (e.g., N1 at 90 ms and P2 at ~170 ms) have been shown to consistently increase in amplitude in response to attended auditory stimuli compared to non-attended stimuli [20,22]. These components are characterized by fronto-central topographies [23,24] and can be captured through around-the-ear EEG electrodes [8]. During the auditory task, participants sat two meters away from the loudspeaker and were instructed to look at a fixation cross in front of them and to listen attentively to the presented tone sequence. No active response to the stimuli was required. Pure tones of 600 Hz lasting 100 ms were repeatedly presented, with a pseudo-random interstimulus interval (ISI) between 1000 and 2000 ms. The task contained 400 tones, lasting approximately 12 min altogether. This task was performed once at the beginning of the experiment and again later in the day.

#### 2.4.2. Impedance Fluctuation Processing

To assess signal quality, changes in impedance values over time were compared for both material types. To this end, the EEG signals were preprocessed by concatenating all the datasets for each participant (i.e., the AEP and visual oddball task recordings, as well as the long-term recording) in the order in which they were recorded. Any package loss caused by the Bluetooth connection or by participants leaving the room was detected using the recorded package number information. A moving rectangular window with a length of 0.5 s and no overlap was then applied to the data. Impedance values were computed for each of these time windows. Some segments contained a large number of invalid impedance values due to movement artifacts, signal packet loss when participants left the room, or electrode contact loss. At this stage of the analysis, we did not distinguish between the different factors causing these values to be invalid. Therefore, we introduced a general cut-off value of 200 kΩ. To conduct a robust linear regression [25] on the continuous impedance data, statistical outliers below the 200 kΩ threshold were removed. The outlier threshold was defined as the median impedance of each participant’s dataset plus half its standard deviation. After applying robust linear regression to each participant’s and channel’s individual impedance data, the respective slopes of the regression lines were calculated.

#### 2.4.3. AEP Processing

The auditory task EEG data were first filtered using a zero-phase, fourth-order Butterworth filter with a stop band between 45 and 55 Hz to remove line noise, and a second stop band between 60 and 65 Hz to remove noise caused by the impedance current. The data were then bandpass-filtered (zero-phase, fourth-order Butterworth) between 0.5 and 20 Hz to remove drift and high-frequency noise. Any signal packet loss caused by the Bluetooth connection issues, e.g., by participants leaving the room, was detected using the recorded packet number information for identification in later processing. The data were segmented into 500 ms epochs beginning 250 ms before and ending 250 ms after stimulus onset. The epochs were then baseline-corrected. Epochs containing signal loss or amplitudes exceeding a threshold of 30 µV (e.g., due to eye blinks) were rejected. To ensure a fair comparison between conditions, the number of epochs was equalized by randomly removing epochs from the measurement, with more trials remaining for each material comparison at the participant level. The average epoch was computed for each subject using a weighted average, whereby each trial was weighted according to the inverse power of its noise to optimize the signal-to-noise ratio (SNR) [26,27]. Additionally, two iteration steps were executed using residual noise from the previous step for improved noise estimation [26,27]. Based on these individual AEP curves, the group average was derived. To estimate the signal quality, the decibel signal-to-noise ratio (dB SNR) was computed for each individual AEP curve using the following equation:$$dB\;SNR=20\;*\;\log_{10}\left(\frac{U_{eff}(Signal)}{U_{eff}(Noise)}\right)$$

The effective potential (U_eff_) of the noise was defined as the root mean square of the averaged epochs from 250 ms before stimulus onset until stimulus onset. The effective potential of the signal was defined as the root mean square (RMS) of the averaged epochs from stimulus onset until 250 ms after stimulus onset. The approach using the evoked potential’s RMS instead of the peak was motivated by SNR calculations in the audio field [27].

### 2.5. Statistical Analysis

The data were statistically analyzed using JASP software v. 0.95.2 [28]. For repeated measures analysis of variance (rANOVA) tests, degrees of freedom and *p*-values were reported using the Greenhouse–Geisser (GG) correction when applicable. Effect sizes for *p* < 0.05 were reported using partial eta squared (ηp^2^). Significant effects were followed up with post hoc paired *t*-tests. In the case of multiple *t*-tests, *p* values were corrected for multiple corrections using the Bonferroni–Holm correction [29,30]. Shapiro–Wilk tests were employed before performing the ANOVA test, to assess normality. For dB SNR, a 2 × 2 rANOVA with the factors of Time (morning vs. afternoon) and Conductive Material (hydrogel vs. silicone) was used. To analyze the slope of the linear regression lines computed for the continuous impedance data, a paired-sample *t*-test (two-tailed) was conducted.

### 2.6. New Materials vs. Standard Electrodes

To better understand how the conductive materials in combination with the flex-printed trEEGrid sensor system compares to standard state-of-the-art EEG recordings, we collected some additional data. We used standard Ag/AgCl ring electrodes and compared them to hydrogel electrodes in the first test and silicone electrodes in the second test. Details of this comparison and its results can be found in the Appendix A.

## 3. Results

### 3.1. Impedance Fluctuation over Time

To assess the quality of the signal, we compared the change in impedance values over time for both material types. First, we examined the change in impedance at the beginning and end of the morning and afternoon measurements. To achieve this, 10 s segments were selected as representative in order to derive average impedance values for all channels and participants. The fluctuations over the entire recording time were also considered for one selected channel. Some channels showed impedance values above 200 kΩ for certain participants in the selected short sections. The affected channels are shown in Figure 5.

For one group of participants, channels with the silicone material were more often affected. In two cases (participants 3 and 6), the impedance was above the threshold of 200 kΩ. In individual cases, some other channels also exceeded the threshold, which occurred more frequently with the silicone material. Some channels only exhibited these issues in the morning, some only at the end of the measurement, and some in both cases. Overall, the hydrogel data of eleven participants and the silicone data of seven participants stayed under the threshold for all channels.

Based on these impedance segments in the morning and afternoon, the boxplot overview presented in Figure 6 was derived, showing the median, minimum and maximum, the 25th and 75th percentiles, and potential outliers. An outlier was defined as a value more than 1.5 times the interquartile range away from the bottom or top of the box. The outliers at the 200 kΩ level are the values of participants whose impedance values exceeded the 200 kΩ threshold, as marked in Figure 6.

For the hydrogel material, the median value for all channels except 1 and 5 was smaller in the afternoon than in the morning. The median values of channels 1 to 5 were all below 73 kΩ. Except for channel 2 in the morning (72.5 kΩ), the values of these channels were all below 61 kΩ. Channels 6 to 8 showed larger impedance values of up to 100.5 kΩ, such as channel 6 in the morning. The median value for channel 8 dropped from 87.3 kΩ in the morning to 55.1 kΩ in the afternoon.

For the silicone material, the percentile range for channels 1 to 3 was narrower than for the hydrogel material, but outlier values were observed as well. The median impedance values for channels 1 to 5 were below 60 kΩ, except for channels 4 (75.8 kΩ) and 5 (88.6 kΩ) in the afternoon. In all channels, the median value increased from the morning to the afternoon; the increase was largest in channel 6, rising from 62.4 kOhm to 118.1 kOhm. Channels 6 and 8 showed greater variance than hydrogel channels. This included outliers above the 200 kOhm threshold. The differences between channel groups 1 to 5 and 6 to 8 may be related to different locations on the face. Channels 6 to 8 were located on the chin (see Figure 2) and were therefore susceptible to movement caused by speaking and chewing. Channel 1, which was located on the forehead, showed consistently less variance compared to other channels, and was therefore selected for further comparison. Furthermore, channel 1 showed a clear AEP response, which further supports the selection of channel 1 for the analyses. Regarding the AEP evaluation, it is noteworthy that the silicone material displays a higher number of valid epochs (in 17 of 19 subjects in the morning session and in 16 of 19 subjects in the afternoon session) compared to the hydrogel material. However, it is important to note that the number of epochs selected for the calculation of the AEPs was adjusted so that an equal number of epochs for both materials were included in the computation (see Section 2.4.3). The maximum difference in epochs between both materials was 45 in the morning and 78 in the afternoon, with an average difference value of 18 in the morning and 27 in the afternoon.

Figure 7 shows the continuous impedance values for each participant and test material, with each dot representing the impedance measured over a 0.5 s window. For better visualization, a maximum cut-off of 200 kΩ was used. Therefore, all values exceeding this threshold due to poor skin contact, participant movement, or disconnection due to the participant leaving the room were set to 200 kΩ. It is noticeable that both conductive materials showed comparable impedance ranges. Further periods of rapidly rising impedance values reaching the maximum cut-off of 200 kΩ are visible. These occurred during recording breaks between tasks, or when participants moved around a lot or left the room (e.g., during the lunch break). Visually, it seems that the impedance of the hydrogel decreased slightly over time, whereas that of the silicone material increased. To investigate this further, we fitted a robust linear regression to the impedance of channel 1 for each participant and test material, discarding outliers, as described in Section 2.4.2.

Based on this data and taking the additional outlier detection into account (see Section 2.4.2), regression lines were derived (see Figure 8). The hydrogel regression lines revealed predominantly negative slopes, while the silicone regression analysis showed predominantly positive slopes. The median slope and its standard deviation for the hydrogel are −3.7 ± 5.4 × 10^−4^, and for the silicone they are 8.8 ± 5.4 × 10^−4^ (see Figure 9). A two-tailed paired-sample *t*-test confirmed that the regression lines of the test materials were significantly different (t(18) = −6.974, *p* < 0.001).

### 3.2. AEP

Figure 9 shows the group-average AEPs for all 19 participants, based on channel 1 of both test materials during the morning and afternoon sessions. Channel 1 was chosen for AEP analyses as the expected AEP response was most clearly pronounced here as opposed to the other channels, due to this channel’s fronto-central location [23,24].

The number of remaining trials available to compute the individual AEPs ranged from 135 to 368 (mean 260) in the morning, and from 129 to 385 (mean 243) in the afternoon. Overall, the group-average AEP morphology and amplitudes were comparable for both test materials and test sessions. The general morphology resembles the standard P1-N1-P2 AEP traditionally measured at the vertex (e.g., [20]). Also, the standard error is similar for both materials.

Figure 10 shows the dB SNR values of each participant, derived from the averaged AEPs recorded for each test material and auditory task. In the morning, hydrogel shows a wider range than silicone. Measurements taken in the afternoon showed a lower median value for hydrogel, indicating poorer SNR compared to silicone. The SNR range is wider for silicone in the afternoon than in the morning.

ANOVA analysis of the dB SNR values revealed a main effect of conductive material (F(1, 18) = 4.96; *p* = 0.039; ηp^2^ = 0.22). There was no significant effect of time (F(1, 18) = 1.45; *p* = 0.24; ηp^2^ = 0.03) or interaction effect (F(1, 18) = 2.94; *p* = 0.10; ηp^2^ = 0.14).

To disentangle the SNR differences between the two materials, two additional rANOVAs were conducted, one on the noise values (U_eff_(Noise)) and one on the signal values (U_eff_(Signal)). ANOVA analysis of (U_eff_(Signal)) revealed no main effect of conductive material (F(1, 18) = 0.27; *p* = 0.61; ηp^2^ = 0.02), no effect of time (F(1, 18) = 0.23; *p* = 0.64; ηp^2^ = 0.01), and no interaction effect (F(1, 18) = 1.02; *p* = 0.33; ηp^2^ = 0.05).

ANOVA analysis of (U_eff_(Noise)) revealed a main effect of conductive material (F(1, 18) = 6.57; *p* = 0.02; ηp^2^ = 0.27). There was no significant effect of time (F(1, 18) = 0.26; *p* = 0.62; ηp^2^ = 0.01) or interaction effect (F(1, 18) = 1.07; *p* = 0.31; ηp^2^ = 0.06).

To further understand the relationship between impedance and SNR, we conducted additional correlational analyses between both parameters. To do that, we extracted the impedance values of both materials during the AEP tasks in the morning and the afternoon from channel 1, as well as the SNR values of the AEP tasks also from channel 1. As the bivariate normality was violated, the non-parametric Spearman’s rho was chosen to compute the correlation between impedance and SNR values at channel 1. The results show no significant correlations between impedance and SNR values (see Table 1).

## 4. Discussion

In this study, we compared two pre-applied conductive materials for flex-printed EEG electrode grids (trEEGrids), which were used as self-applicable electrode patches during five- to six-hour recording sessions. We evaluated the signal quality based on impedance levels and fluctuations over time, as well as on AEP (P1-N1-P2 complex), pre-stimulus noise levels, and SNR values derived from AEP curves.

Impedance measurements for both conductive materials were found to be within a comparable range. However, compared to the hydrogel material, the silicone material showed a higher outlier rate, exceeding the 200 kΩ threshold for specific electrodes and participants, based on the first and last 10 s of the measurement. At the same time, the silicone material showed comparable performance to the hydrogel material for other electrodes and participants. It is difficult to clearly identify the reasons for the outliers as various factors come into play. Careful attention was paid to ensuring that the skin was prepared in the same manner; however, some differences in preparation cannot be ruled out entirely. The electrode patches were also manufactured manually, so variations are possible.

Overall, the impedance level for both materials is much higher than that of standard laboratory ring electrodes, which have impedance levels of a few kΩ. However, the reported impedance levels are lower than those achieved by dry electrode systems [3], which are comparable in terms of self-application. Notably, regression analysis indicated a slight decrease in impedance for the hydrogel material, while a slight increase was observed for the silicone material over the recording period. This decrease in the hydrogel material may be due to its moisturizing effect on the skin. While this is beneficial for the impedance level, it could also put additional strain on the skin with continuous and repeated usage. In contrast, the silicone material does not exhibit this moisturizing effect due to its dry nature.

For practical application, silicone electrodes have the advantage of not drying out, potentially resulting in a longer shelf life without the need to store them in sealed packages. It is also worth noting that the spherical shape of the hydrogel lenses was not optimally suited to achieve a larger contact surface area. Without adhesive to hold it in place, the contact area on the skin was reduced, resulting in poorer impedance levels. These aspects mark room for improvement for future work using a flatter design.

It is also noteworthy that both materials are sensitive to handling. While the hydrogel material appeared reasonably robust, its surface should only be placed on cleaned skin to avoid contamination from any remaining sebum. The silicone material on the other hand is more robust in this respect, as it hardly absorbs any contamination. However, it is deformable, which makes it challenging to place on the electrode surface with good contact over a large area. Both versions of the electrode patch produced reasonable results, as they captured AEPs with acceptable and comparable SNR values. However, establishing a ground truth in the EEG range is challenging due to various factors, including skin type, skin preparation, and individual AEP characteristics [31]. This is also an explanation for the lack of a clear relationship between impedance values and SNR reported in this study. Both parameters are relevant and important evaluation criteria but just looking at impedance is not sufficient to explain differences in SNR. Other factors, such as skin type, skin preparation, and individual AEP characteristics such as neural source location and orientation, contribute as well. The issue of impedance is of relevance as an evaluation criterion, particularly in the context of mobile EEG measurements. The reason for this is that high impedance values increase susceptibility and sensitivity to external noise and artefactual noise [32]. Since the AEP measurement was conducted in a lab environment where participants sat still, good AEP signals were obtained even when impedance values were high. Therefore, future studies should more closely investigate the impact of impedance values on EEG signals in real-world environments and activities.

A limitation of this study is the absence of traditional electrodes for comparison. Instead, we focused on a direct comparison between silicone and hydrogel materials. Future studies should incorporate both materials alongside Ag/AgCl rigid electrodes as the gold standard, to better identify potential drawbacks in signal quality that may not have been observed here. However, to address this limitation to a certain extent we collected a small dataset comparing the new materials to standard Ag/AgCl ring electrodes. The results of this suggest that the EEG data obtained with the new materials are comparable to state-of-the-art laboratory EEG electrodes (S1).

Hydrogels can swell due to fluid absorption [33]; however, we did not observe significant swelling in this study. This is likely because the study was carried out in a dry environment in the wintertime, when cooler temperatures do not cause excessive sweating, etc. Therefore, we did not measure whether this hydrogel swelling might have impacted the impedance levels.

Hydrogel electrodes have a natural tendency to lose moisture and dry out, which could affect their conductive properties and consequently the signal. Although we mitigated this problem by using circular adhesives around the hydrogel material to seal it off, loss of moisture cannot be ruled out as we did not specifically measure the water content of the hydrogel material before and after the measurements. This should be addressed in future studies, especially for measurements exceeding six hours. Moreover, field studies in real-world settings are essential to advance mobile EEG technology—moving it from primarily clinical and laboratory applications toward applied, assistive technologies. We believe that flex-printed electrode technology represents an important step toward achieving this goal.

## 5. Conclusions

In conclusion, both hydrogel and silicone pre-applied conductive materials on a trEEGrid patch demonstrated promising performance for EEG measurements lasting five to six hours. These solutions provide a foundation for developing sensor systems that are self-applicable and comfortable for extended wear. This opens up new possibilities for mobile EEG data collection, spanning medical applications (e.g., sleep and epilepsy monitoring), mental state assessments, and neuropsychological research, including home-based monitoring scenarios. Future work will focus on further development and evaluation of these technologies for use in everyday life.

## Figures and Tables

**Figure 1 sensors-25-06810-f001:**
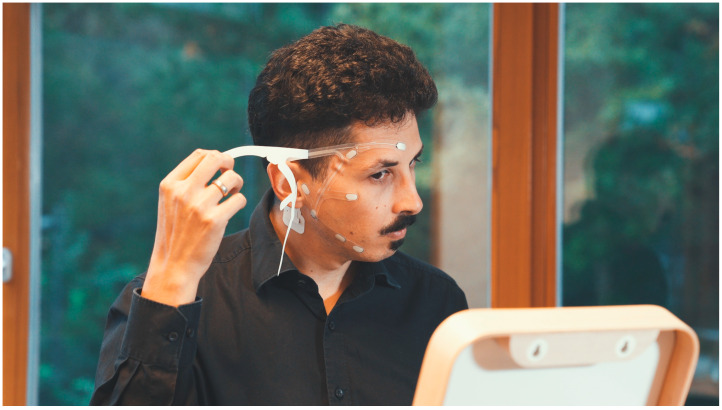
Self-application of trEEGrid electrode patch. ©Fraunhofer IDMT/Leona Hofmann.

**Figure 2 sensors-25-06810-f002:**
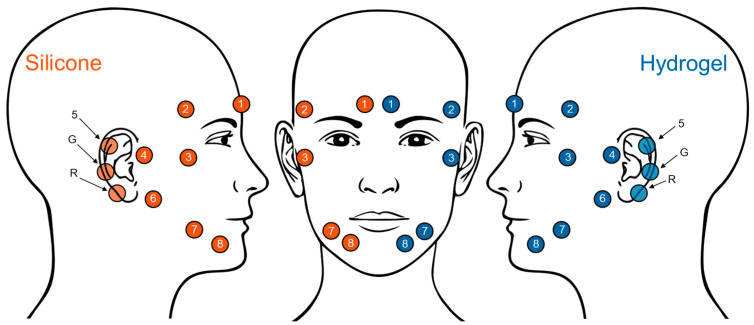
Channel placement for the two material types, with electrode channels using the silicone-based conductive material (orange) placed on the right part of the head and those using the hydrogel conductive material (blue) placed on the left part of the head. Electrodes 5, ground (G), and reference (R) are located behind the ear.

**Figure 3 sensors-25-06810-f003:**
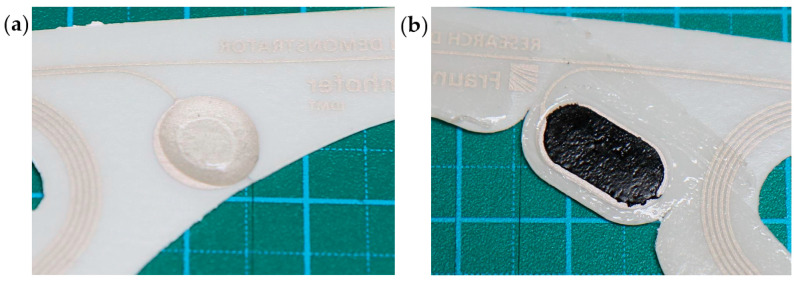
Demonstration of both conductive materials used in this study: (**a**) hydrogel: solid gel based on hydrogel electrolyte; (**b**) silicone: silicone-based, soft-skin conductive tape. Regarding the dimension, each solid square measures 10 × 10 mm and each dotted square 5 × 5 mm.

**Figure 4 sensors-25-06810-f004:**
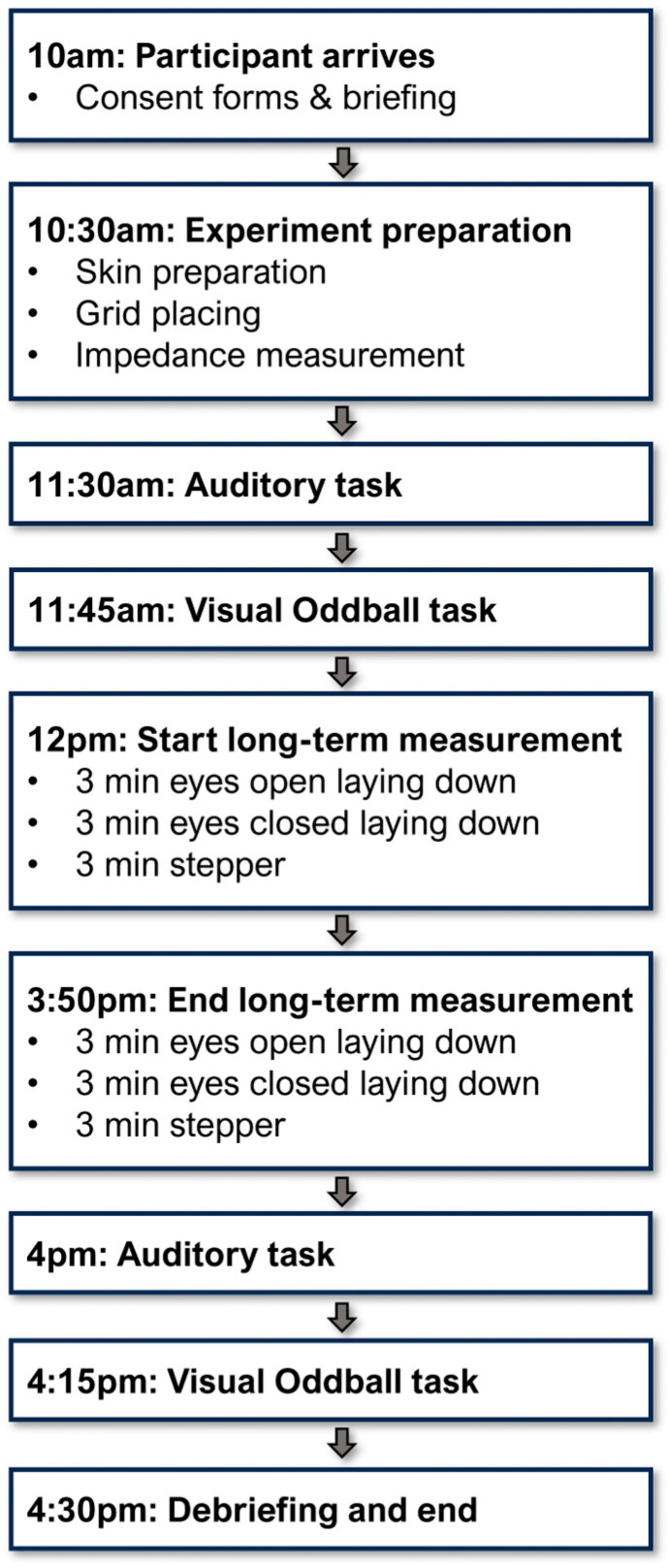
Measurement protocol throughout the day, with the approximate start and end times for each task component. Continuous impedance measurement took place from the beginning of the first impedance measurement (~10:30 a.m.) to the end of the visual oddball task (~4:30 p.m.) by concatenating the subsequent data files. Visual oddball tasks, as well as eyes-open, eyes-closed and stepping tasks, were not analyzed further for this study.

**Figure 5 sensors-25-06810-f005:**
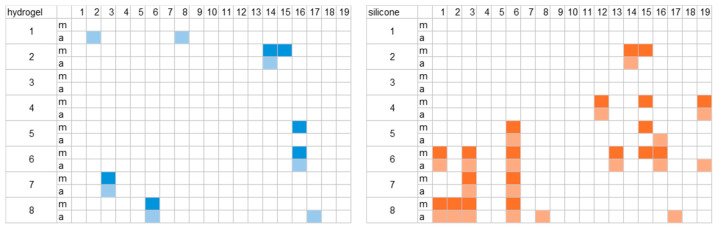
Overview of channels with impedance values exceeding the threshold of 200 kΩ for the entire duration of analysis (10 s) in the morning (m) and in the afternoon (a). Rows indicate channels 1 to 8’ the columns indicate participants 1 to 19. The results for the hydrogel materials are presented on the **left** (blue); the results for the silicone materials are presented on the **right** (orange).

**Figure 6 sensors-25-06810-f006:**
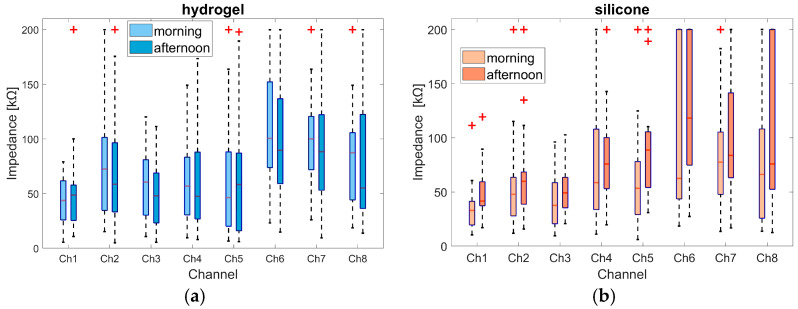
Comparison of the impedance values between the morning and the afternoon for all channels across participants, calculated separately for (**a**) hydrogel and (**b**) silicone. Each boxplot shows the median value as a red line; the ends of the colored boxes represent the 25th and 75th percentiles; minimum and maximum impedance values are marked at the ends of the dotted black lines. Outliers, identified based on results 1.5 times the interquartile range away from the bottom or top of the box, are marked as red crosses.

**Figure 7 sensors-25-06810-f007:**
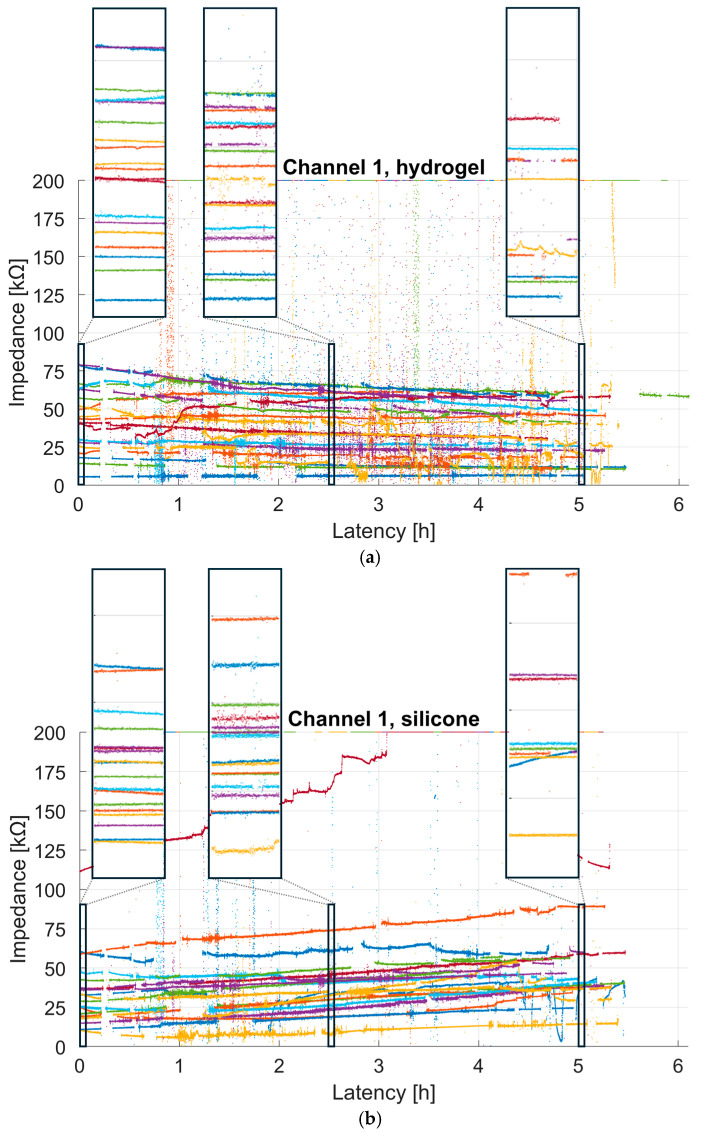
Progression of impedance values in 0.5 s segments over the course of the day (5.5 h) of channel 1 for each participant and test material: (**a**) hydrogel and (**b**) silicone. A maximum cut-off of 200 kΩ was used, as described in Section 2.4.2. Each color represents one participant. Each of the three marked sections show a 180 s excerpt from the start (0 h), middle (2.5 h), and end (5 h) of the impedance curves for hydrogel and silicone.

**Figure 8 sensors-25-06810-f008:**
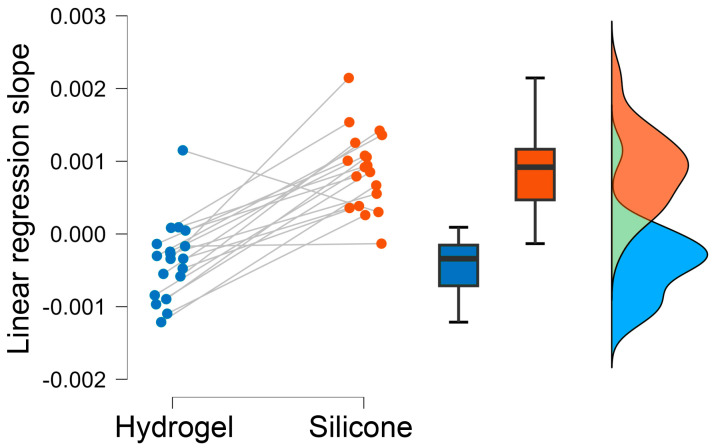
Depiction of individual linear regression slope values (not normalized) for hydrogel (blue) and silicone (orange) to visualize the impedance fluctuations over time for both materials. Boxplots of the regression slopes (middle) show the median value, the first and third percentiles, and minimum and maximum values, excluding outliers exceeding a value of 1.5 times the inter-quartile range from the first and third quartile. Distribution curves (right) are depicted as well. The green color indicates the overlap of both distributions.

**Figure 9 sensors-25-06810-f009:**
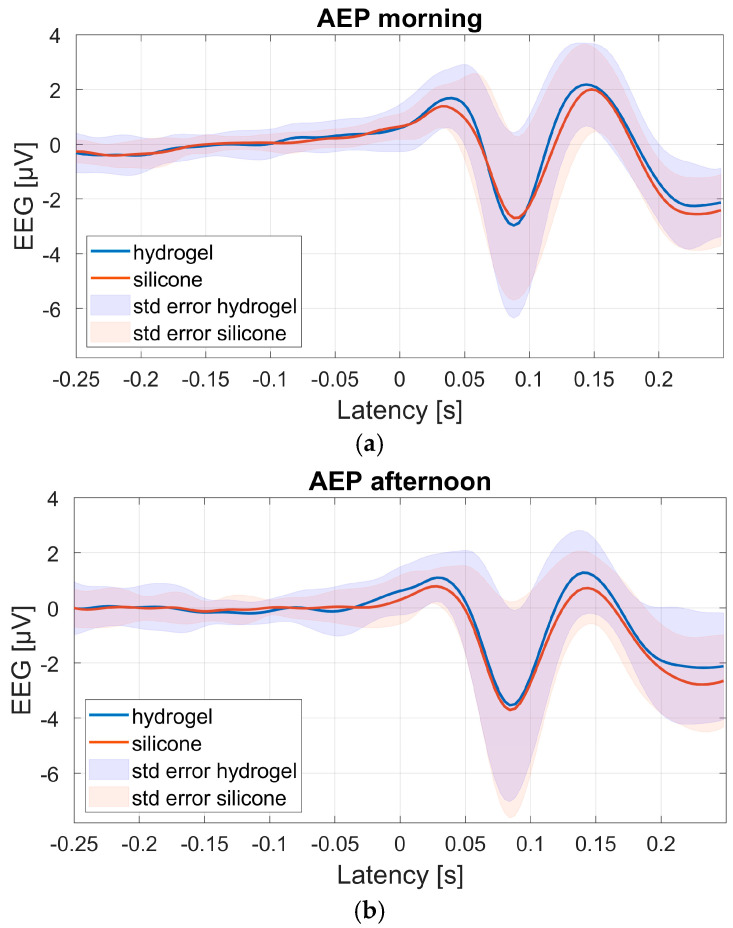
Group average auditory task AEP at channel 1 in the morning (**a**) and afternoon (**b**) sessions. The standard error is shown as the shaded area.

**Figure 10 sensors-25-06810-f010:**
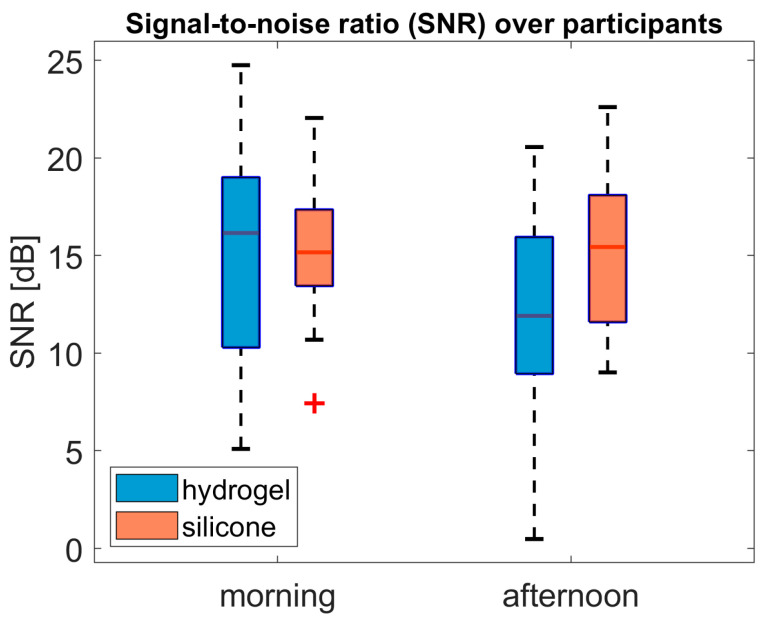
Individual dB SNR values for each participant and test material, showing the change over time for channel 1 with hydrogel and silicone. The boxplot shows the median value as red line, the ends of the colored boxes represent the 25th and 75th percentiles, and minimum and maximum values are marked at the ends of the dotted black lines. Outlier values more than 1.5 times removed from the top or bottom interquartile border (bottom or top of box) are marked as red cross.

**Table 1 sensors-25-06810-t001:** This table contains the results of Spearman’s rho correlational analyses between impedance and SNR values collected at channel 1 from both materials during the AEP measurement in the morning and the afternoon.

Correlation: Impedance and SNR	Morning (rho)	Afternoon (rho)
Hydrogel	−0.03; *p* = 0.91	0.07; *p* = 0.77
Silicone	−0.32; *p* = 0.18	−0.46; *p* = 0.05

## Data Availability

The raw data supporting the conclusions of this article will be made available by the authors, upon request and in accordance with the consent of the participants as per the data protection agreement.

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
