# Peer review of "Evaluation of Pre-Applied Conductive Materials in Electrode Grids for Longterm EEG Recording"

_sensors, 2025, doi:10.3390/s25226810_

Round 1

Reviewer 1 Report

Comments and Suggestions for Authors

In the manuscript entitled “Evaluation of Pre-Applied Conductive Materials in Electrode Grids for Long-Term EEG Recording”, the authors present a flexible EEG grid incorporating pre-applied conductive materials designed to record auditory evoked potentials (AEPs). The principal contributions of this work are: (1) comparison of two conductive materials (hydrogel and silicone) integrated into a flexible EEG grid; (2) impedance characterization of hydrogel- and silicone-based electrodes; and (3) demonstration of AEP recordings obtained using the developed EEG grid.

This study introduces a potentially valuable approach for long-term EEG acquisition, offering the advantage of self-application without the need for additional gel treatment. The work is well-motivated and thoughtfully designed; however, the validation requires further clarification and strengthening. Therefore, I recommend the manuscript be considered for publication in Sensors pending major revision and resolution of the following issues:

  1. The authors have previously published on trEEGrid (Souto et al., Front. Neurosci., 2022). The introduction should explicitly clarify the novel contributions of the present study and highlight the advantages of this work over their earlier publication.
  2. The manuscript should specify which hydrogel material (e.g., PVA, PAA, PU) was employed. Given that hydrogels can dehydrate during extended EEG procedures, the authors need to describe any strategies used to mitigate drying and maintain material stability during recordings of up to 6 hours.
  3. In section 3.1 (lines 306-309, page 8), hydrogels can absorb fluids such as sweat or body fluids during contact with the skin. The authors need to discuss whether such swelling could cause impedance fluctuations and how this might impact signal-to-noise ratio (SNR) in AEP recordings.
  4. Figure 4 presents three experimental tasks (eyes open lying down, eyes closed lying down, and stepper). However, Figure 9 includes only a single AEP result. The manuscript should clarify which task this result corresponds to. In addition, I suggest demonstrating the device’s ability to capture and distinguish AEPs across the different tasks.
  5. The relationship between electrode impedance and SNR should be analyzed more systematically. For instance, does lower electrode impedance correspond to lower SNR in AEP recordings? Moreover, the SNR results in Figure 9 are limited to a single channel, which limits the generalization of the overall SNR results. I recommend including results from at least three different channels, with correlation to impedance values, to provide a more comprehensive validation.

Author Response

  1. The authors have previously published on trEEGrid (Souto et al., Front. Neurosci., 2022). The introduction should explicitly clarify the novel contributions of the present study and highlight the advantages of this work over their earlier publication.
    • Reply: Thank you for pointing this out. We have now provided a specific statement about the contributions of this work in the introduction (on page 3) to underline the novelty of this study to set it apart from the previous studies.
  2. The manuscript should specify which hydrogel material (e.g., PVA, PAA, PU) was employed. Given that hydrogels can dehydrate during extended EEG procedures, the authors need to describe any strategies used to mitigate drying and maintain material stability during recordings of up to 6 hours.
    • Reply: Thank you for pointing this out. We added some additional information on pages 3 and 4. The hydrogel electrodes were surrounded by circular adhesives for two reasons: 1. To ensure adhesion of patch and electrodes for the entire duration of the experiment and 2. To prevent electrodes from drying out. By applying circular adhesives around the hydrogel electrodes, the material was essentially sealed and exposure to air was mitigated. However, we did not directly measure the water content of the hydrogel material before and after. We added this in the limitation on page 13.
  1. In section 3.1 (lines 306-309, page 8), hydrogels can absorb fluids such as sweat or body fluids during contact with the skin. The authors need to discuss whether such swelling could cause impedance fluctuations and how this might impact signal-to-noise ratio (SNR) in AEP recordings.
    • Reply: Thank you for pointing this out. We did not observe any noticeable swelling in the hydrogels throughout the experiment, likely because the study took place in the wintertime and participants were not sweating or exposed to high humidity, etc. Therefore, we did not measure the amount of swelling that may have impacted impedances during this experiment. We have now added a statement on this in the limitations (p. 14).
  1. Figure 4 presents three experimental tasks (eyes open lying down, eyes closed lying down, and stepper). However, Figure 9 includes only a single AEP result. The manuscript should clarify which task this result corresponds to. In addition, I suggest demonstrating the device’s ability to capture and distinguish AEPs across the different tasks.
    • Reply: Thank you for pointing out the lack of clarity when it comes to the auditory task and the AEP’s used for analysis. The AEP signal comes from auditory stimuli and are only from the Auditory task. Data from other tasks were not presented directly in this study (except for impedance data), to stick to a main story and improve readability. We have now clarified this on P. 5, in the Figure 9 caption, as well as throughout the text when referring to the AEP data.
  2. The relationship between electrode impedance and SNR should be analyzed more systematically. For instance, does lower electrode impedance correspond to lower SNR in AEP recordings? Moreover, the SNR results in Figure 9 are limited to a single channel, which limits the generalization of the overall SNR results. I recommend including results from at least three different channels, with correlation to impedance values, to provide a more comprehensive validation.
    • Reply: Thanks for addressing this important point. Channel 1 was chosen for AEP analyses as the expected AEP response was most clearly pronounced here as opposed to the other channels, due to this channel’s fronto-central location [22, 23]. We added this sentence on page 11. This aspect is also relevant for the SNR analyses. We believe that an SNR analysis is warranted only for channels where a clear signal – in this case a typical AEP (P1-N1-P2 complex) – is observable. Therefore, we restricted the SNR analyses to data from channel 1. To address the relationship between SNR and impedance we conducted a correlational analysis of both parameters for channel 1. We added the results of this analysis on page 12. No significant correlation was found between SNR and impedance. We address this relationship further in the Discussion on page 14. Both parameters are relevant and important evaluation criteria but just looking at impedance is not sufficient to explain differences in SNR. Other factors such as skin type, skin preparation and individual AEP characteristics such as neural source location and orientation contribute as well. However, impedance is nevertheless an important parameter to measure and to be used as evaluation criterion particularly in the context of mobile EEG measurements. The reason being that high impedance values increase the susceptibility and sensitivity to external noise and artefactual noise. Since the AEP measurement was conducted in a lab environment where participants sat still, good AEP signals were obtained even when impedance values were high.

Reviewer 2 Report

Comments and Suggestions for Authors

Strengths:

  • The authors have used appropriate testing and tools with good demonstration of analytical clarity, starting with the symmetric layout of the electrodes that enables intra-subject comparison of both materials
  • Detailed descriptions of materials, methods and protocol, enhancing the reproducibility of the study
  • Robust, descriptive use of statistical tests

Weaknesses:

  • While mentioned by the authors, the lack of a comparison to the gold standard in electrodes [Ag/AgCl] could help any claims of equivalence or superiority of the trEEGrids when used w/ either application materials
  • Although addressed as a limitation, it would serve the audience better if a comparison [even minimalistic] to traditional electrodes is presented; This could help address the importance beyond usability and comfort

Author Response

  1. While mentioned by the authors, the lack of a comparison to the gold standard in electrodes [Ag/AgCl] could help any claims of equivalence or superiority of the trEEGrids when used w/ either application materials
  2. Although addressed as a limitation, it would serve the audience better if a comparison [even minimalistic] to traditional electrodes is presented; This could help address the importance beyond usability and comfort
    • Reply: We agree with the reviewer of the limitation of not including SoTA electrode systems in the study. We have done such a comparison between trEEGrid and cap abralyt electrodes in previous studies for spectral activity (Scanlon et al., 2025) and sleep monitoring (da Silva Souto et al., 2022) and have now mentioned this in the introduction (p.3).
    • Reply: To address the difference between the new materials used in this study and SoTA electrodes, we added a supplementary materials section (S1) in which we show and discuss the results of a comparison test between new materials and Ag/AgCl ring electrodes. We refer to this section at the end of the Methods section (2.6) and in the discussion on page 13. Despite the limited size of the test data, we hope to have addressed your valid comment.

Reviewer 3 Report

Comments and Suggestions for Authors

The article describes very promicing results in the development of the novel type of electrodes for long-term EEG measurements. The main problem of this article, that require the major revision is the quality of the Figures:

  1. Image quality of all figures is low - it contains too thin lines, unreadable text and similar kind of problems
  2. it is better to describe figures and subplots not by their type (boxplot overview, linear regression solope), but by the dependencies of physical parameters they should show.
  3. It is better to rearrange the figures into the raw values recordings at several timescales (I would also suggest to add comparison against the usual state-of-the-art EEG electrodes on shorter timescale at least once) - hour, minute, short period; then some data statistics, then the proper approximations deriving some physical values.
  4. I would also suggest to extend the comparison agianst the SoTA sensors a bit - showing what data degradation appears(if any) comparing to the currently used EEG electrodes.

Author Response

  1. Image quality of all figures is low - it contains too thin lines, unreadable text and similar kind of problems
    • Reply: Thank you for this feedback. We have now worked on the figure quality for figures 5, 6, 7, 9 and 10 regarding font size, line thickness, proportions.

2. it is better to describe figures and subplots not by their type (boxplot overview, linear regression solope), but by the dependencies of physical parameters they should show.

    • Reply: Thanks for pointing this out. We have adjusted the figure captions (figures 6, 7, 8) to describe more clearly what they depict.

3. It is better to rearrange the figures into the raw values recordings at several timescales (I would also suggest to add comparison against the usual state-of-the-art EEG electrodes on shorter timescale at least once) - hour, minute, short period; then some data statistics, then the proper approximations deriving some physical values.

    • Reply: Thank you for this comment. In reference to the SoTA electrodes, refer to our response to comment #4 below as it relates to the same issue.

4. I would also suggest to extend the comparison agianst the SoTA sensors a bit - showing what data degradation appears(if any) comparing to the currently used EEG electrodes.

    • Reply: We agree with the reviewer of the limitation of not including SoTA electrode systems in the study. We have done such a comparison between trEEGrid and cap abralyt electrodes in previous studies for spectral activity (Scanlon et al., 2025) and sleep monitoring (da Silva Souto et al., 2022) and have now mentioned this in the introduction (p.3).
    • To address the difference between the new materials used in this study and SoTA electrodes, we added a supplementary materials section (S1) in which we show and discuss the results of a comparison test between new materials and Ag/AgCl ring electrodes. We refer to this section at the end of the Methods section (2.6) and in the discussion on page 13. Despite the limited size of the test data, we hope to have addressed your valid comment.

Round 2

Reviewer 1 Report

Comments and Suggestions for Authors

The authors have satisfactorily addressed all of the reviewer’s concerns, and the manuscript is now suitable for publication in Sensors.

Author Response

Dear Reviewer, thanks for your feedback throughout the review process and for your recommendation to publish.

Reviewer 3 Report

Comments and Suggestions for Authors

I would still suggest to extend Figure 4 description a bit making it fully descriptive without main text of the article. I also suggest to rework Fig.7 content - it is still too cluttered and it can be made more informative by providing less raw data and working with several timescales - for example it can be extended with raw data at other timescales than the provided raw data.

Author Response

Dear Reviewer,

thanks for your recommendations.

Comment 1: I would still suggest to extend Figure 4 description a bit making it fully descriptive without main text of the article. 

Reply: We have added a more informative description to figure 4: "Figure 4: Protocol of measurement throughout the day, with the approximate start and end times for each task component. Continuous impedance measurement took place from the beginning of the first impedance measurement (~10:30am) to the end of the visual oddball task (~4:30pm), by concatenating the subsequent data files. Visual oddball tasks, as well as eyes open, eyes closed and stepping tasks were not analysed further for this study."    

Comment 2: I also suggest to rework Fig.7 content 

Reply: "Thank you for the suggestion to improve the figure. We have revised figure 7 by showing data from shorter time segements to provide a more detailed insight into the impedance data and how it progresses over time"